# Neoadjuvant Chemotherapy Followed by Radiofrequency Ablation May Be a New Treatment Modality for Colorectal Liver Metastasis: A Propensity Score Matching Comparative Study

**DOI:** 10.3390/cancers14215320

**Published:** 2022-10-28

**Authors:** Yizhen Chen, Yurun Huang, Linwei Xu, Jia Wu, Fang Han, Hang Jiang, Pengwen Zheng, Dong Xu, Yuhua Zhang

**Affiliations:** 1Department of Hepatobiliary and Pancreatic Surgery, The Cancer Hospital of the University of Chinese Academy of Sciences (Zhejiang Cancer Hospital), Institute of Basic Medicine and Cancer (IBMC), Chinese Academy of Sciences, Hangzhou 310022, China; 2Zhejiang Chinese Medical University, Hangzhou 310053, China; 3Department of Ultrasound, The Cancer Hospital of the University of Chinese Academy of Sciences (Zhejiang Cancer Hospital), Institute of Basic Medicine and Cancer (IBMC), Chinese Academy of Sciences, Hangzhou 310022, China

**Keywords:** colorectal cancer, liver metastasis, neoadjuvant chemotherapy, radiofrequency ablation, liver resection

## Abstract

**Simple Summary:**

Whether neoadjuvant chemotherapy followed by radiofrequency ablation, a new therapeutic modality, may achieve a similar prognosis to neoadjuvant chemotherapy followed by hepatectomy has yet to be explored. A total of 190 colorectal liver metastases patients met the inclusion criteria. The 3-year progression-free survival in the neoadjuvant chemotherapy followed by liver resection and neoadjuvant chemotherapy followed by the radiofrequency ablation groups was 38.8% vs. 55.3%. Compared with the neoadjuvant chemotherapy followed by hepatectomy, neoadjuvant chemotherapy followed by radiofrequency ablation has the characteristics of rapid recovery, lower progression rate, and fewer complications.

**Abstract:**

Background: Most colorectal liver metastases (CRLM) are not candidates for liver resection. Radiofrequency ablation (RFA) plays a key role in selected CRLM patients. Neoadjuvant chemotherapy (NAC) followed by liver resection has been widely used for resectable CRLM. Whether NAC followed by radiofrequency ablation (RFA) can achieve a similar prognosis to NAC followed by hepatectomy remains is unclear. The present study aimed to provide a new treatment modality for CRLM patients. Methods: This comparative retrospective research selected CRLM patients from 2009 to 2022. They were divided into NAC + RFA group and NAC + hepatectomy group. The propensity score matching (PSM) was used to reduce bias. We used multivariate cox proportional hazards regression analysis to explore independent factors affecting prognosis. The primary study endpoint was the difference in the progression-free survival (PFS) between the two groups. Results: A total of 190 locally curable CRLM patients were in line with the inclusion criteria. A slight bias was detected in the comparison of basic clinical characteristics between the two groups. RFA showed a significant advantage in the length of hospital stay (median; 2 days vs. 7 days; *p* < 0.001). The 1- and 3-year PFS in the liver resection and the RFA groups was 57.4% vs. 86.9% (*p* < 0.001) and 38.8% vs. 55.3% (*p* = 0.035), respectively. The 1-year and 3-year OS in the liver resection and RFA groups was 100% vs. 96.7% (*p* = 0.191) and 73.8% vs. 73.6% (*p* = 0.660), respectively. Conclusions: NAC followed by RFA has rapid postoperative recovery, fewer complications, and better prognosis.

## 1. Introduction

Colorectal cancer (CRC) is one of the cancers with the highest mortality and morbidity worldwide [1]. Liver metastases are the most frequent sites of distal spread in CRC. About 50% of CRC patients develop liver metastasis during the disease, which worsens the overall prognosis of patients [2,3,4].

Liver resection was considered the only curative treatment modality for initially locally curable colorectal liver metastasis (CRLM) [5,6]. CRLM patients have a 5-year survival rate in 50% after liver resection, which significantly prolongs their overall survival (OS) [6,7]. Due to the high rate of metastatic recurrence in CRLM, neoadjuvant chemotherapy (NAC) is routinely used to prolong progression-free survival (PFS) [8,9]. Current guidelines and consensus (including NCCN and ESMO) recommend NAC combined with liver resection as first-line therapy for resectable CRLM [10,11,12,13].

Although the indications of liver resection have expanded, approximately 80% of CRLM patients are not suitable for surgery [4,14]. Local ablation, represented by radiofrequency ablation (RFA), plays a critical role in specific CRLM patients [15,16]. However, for CRLM patients, RFA alone has a worse prognosis than liver resection alone [17,18,19,20].

NAC followed by RFA has been shown to have a better prognosis than RFA alone in locally ablatable CRLM patients [21,22]. Whether NAC plus RFA has a better prognosis than NAC plus hepatectomy is currently under intensive focus. Therefore, this study aimed to identify the optimal treatment modality for initially locally curable CRLM patients.

## 2. Materials and Methods

### 2.1. Population and Grouping

Herein, this research performed the retrospective comparative analysis for CRLM patients. All liver metastases were confirmed pathologically as CRLM by liver biopsy during the treatment. Imaging data of these patients were reassessed by a multidisciplinary team (MDT) prior to NAC. All cases underwent NAC (followed by hepatectomy or RFA) at Zhejiang Cancer Hospital from January 2009 to January 2022. These initially locally curable CRLM patients undergo NAC and locally curative treatment (liver resection or RFA only). According to the results of the MDT, locally curable is considered to refer to the following three points: liver resection or RFA is technically feasible; the patient’s physical condition meets the surgical requirements (the performance status of all CRLM patients is 0–1); it was ensured that all operating requirements were met in accordance with guidelines and consensus [10,11,23], and patients provided informed consent. The primary sites of all cases had been radically resected. The first choice in the NAC followed by RFA group was not ablation (but surgical resection) in this study. The MDT and the patients negotiated to choose RFA for various reasons after NAC.

Eligible CRLM patients were automatically divided into two groups based on the choice after NAC. NAC followed by RFA was the RFA group. NAC followed by hepatectomy was the liver resection group. The liver metastases in the RFA group were completely ablated, and the ablation edge was >10 mm. The liver metastases in the liver resection group were completely removed. We screened out patients, who met any of the following criteria. (1) Initially unsuitable for local curative treatment (radiofrequency ablation and hepatectomy) based on assessment; (2) extrahepatic metastasis; (3) prior to NAC, other treatments, such as hepatic arterial infusion chemotherapy and microwave ablation were performed; (4) severe dysfunction of vital organs; (5) the follow-up was for <6 months; or (6) recurrent liver metastasis (Figure 1). All CRLM patients provided written informed consent. To ensure the accuracy or completeness of this research, any concerns were investigated and appropriately addressed. The authors are responsible for all aspects of the research. The Medical Ethics Committee of Zhejiang Cancer Hospital approved this study. The ethics approval number is IRB-2021-279. The study conformed to the provisions of the Declaration of Helsinki (as revised in 2013).

### 2.2. Surgical Procedure

Ultrasound (US)-guided percutaneous RFA in the RFA group was performed by interventional radiologists and hepatobiliary surgeons. Percutaneous RFA was conducted under general anesthesia. The interventional radiologist had >10 years of experience in percutaneous liver ultrasonography (US) and RFA. The RFA was performed with a 16-G bipolar electrode needle three times per lesion site until the rolling endpoint was reached. The electrodes were placed within the lesion under US guidance. Consequently, an ablation margin of >1 cm was achieved. For lesions >3 cm, multiple overlapping ablations were required, while continuous monitoring of local temperature and tissue impedance was required. During RFA, US in the ablation zone showed a high echo. All patients underwent CT immediately after RFA (to determine if ablation is complete, or whether there was tumor residue).

The chief surgeon with more than 10 years of experience performed laparotomy or minimally invasive liver resection. Intraoperative ultrasound combined with preoperative imaging data was routinely used. We avoided missing liver metastases during surgery. Liver parenchyma-preserving approach surgical resection (surgical margin >1 mm) was used during the operation, and the use of intermittent Pringle maneuver was at the discretion of the surgeon. Subsequently, all liver metastases should be removed. Any other treatments, such as radiofrequency ablation, should not be used during hepatectomy. Before closing the abdominal cavity, US was repeated to avoid omission.

### 2.3. Definitions

The primary endpoint of this study was PFS. PFS is from RFA or liver resection to confirmation of death and recurrence. The secondary endpoints included OS and postoperative complications. NAC was defined as receiving chemotherapy according to the MDT specified regimen before local cure treatment. The effectiveness of RFA is defined as the ablation defect surrounding the target liver metastases, while the failure is defined as evidence of residual tumor within 1 cm of the ablation defect. Local tumor progression (LTP) is defined as a new nodular enhancement or enlargement of ablation defect within 1 cm of the operation area [24]. LTP-free survival (LTPFS) is defined as the duration interval between the first RFA and the occurrence of LTP. Clinical risk score (CRS) has the following indicators: lymph node positive of CRC; more than one liver metastases; diagnosis of liver metastases is less than 1 year from CRC surgery; carcinoembryonic antigen (CEA) >200 (ug/L); the size of the largest liver metastases was >50 mm [10,11]. One point was assigned to each item. The objective efficacy of this research was assessed with reference to RECIST version 1.1 [25].

### 2.4. Data Collection and Follow-Up

The baseline, hospital information (such as hospital stay, postoperative complications, etc.), and NAC regimen of CRLM patients were collected. The follow-up was until 1 January 2022 to obtain the survival status of CRLM patients. The follow-up protocol was in accordance with the guidelines and consensus. Each CRLM patient went to the hospital every 3 months after treatment. Tumor progression was confirmed by MRI and contrast-enhanced US and CT. If the CRLM patient was progression-free in the initial 2 years, the follow-up interval was set to 6 months. If disease-free status was maintained over 5 years, follow-up frequency was adjusted to annual.

### 2.5. Statistical Analysis

Fisher’s exact test, continuity correction, and Pearson’s c^2^ test were used to compare the baseline characteristics. Survival information such as OS or PFS were evaluated by the Kaplan–Meier method. First, the Cox proportional hazards multiple regression model was used for univariate analysis. The different characteristics (*p* < 0.1) in univariate analysis were placed into multivariate analysis, and finally factors with *p* < 0.05 were an independent prognostic factor.

Confounding factors between the two groups need to be addressed. Propensity score matching (PSM) analysis was used to simulate randomization in prospective trials. In this study, multivariate logistic regression was used to measure each propensity score. Independent prognostic factors including gender, T stage of primary tumor, the timing of metastasis, and the largest diameter of liver metastasis were put into the analysis software for matching. Then, the two groups were formed using a one-to-one nearest neighbor caliper with a width of 0.03. All statistical analyses were conducted using SPSS statistical software (version 25, SPSS Inc., Chicago, IL, USA).

## 3. Results

### 3.1. Clinicopathological Characteristics

From 2009 to 2022, all 190 CRLM patients met the inclusion criteria: 61 were NAC followed by RFA and 129 were NAC followed by liver resection group. The clinical characteristics of the two groups are summarized in Table 1. The median age of patients in the RFA and liver resection groups was 57.0 and 58.0 years, respectively. The female:male ratio in the whole cohort was 59:131. There were statistical differences between the two groups in 4 baseline characteristics. In the liver resection group, 57.4% (74/129) of patients used the XELOX regimen, 21.7% (28/129) used the FOLFOX regimen, and 20.9% (27/129) used the FOLFIRI regimen. In the RFA group, 65.6% (40/61) of patients used the XELOX regimen, 16.4% (10/61) used the FOLFOX regimen, and 18.0% (11/61) used the FOLFIRI regimen. Statistically, there was no difference between the two groups (*p* = 0.542). Moreover, 21.3% (13/61) of patients in the RFA group and 31.8% (41/129) in the liver resection group were combined with targeted drugs. The median number of cycles of NAC in the RFA and liver resection groups is 4 (3, 6) and 5 (3, 7). The performance status of all patients was 0–1. About 75% of patients (95/129) in the liver resection group underwent laparotomy. In this study, after PSM, 48 CRLM patients in each group were obtained (Table 2). In matched cohort, no significant differences in any key confounders were found at baseline between two groups. After PSM, there was also no difference in chemotherapy regimen between the two groups (*p* = 0.338). In the liver resection group, 24/48 patients used the XELOX regimen,13/48 used the FOLFOX regimen, and 11/48 used the FOLFIRI regimen. In the RFA group, 31/48 patients used the XELOX regimen, 10/48 used the FOLFOX regimen, and 7/48 used the FOLFIRI regimen.

### 3.2. Perioperative Period

The perioperative analysis of NAC followed by local curative treatment is shown in Table 3. The rate of postoperative complications in the NAC plus liver resection was nearly twice the NAC plus RFA (21.74% vs. 14.8%, *p* = 0.297), although not significantly. The highest incidence of postoperative complication in the liver resection group was pleural effusion, while that in the RFA group was an abdominal infection. Although there was no statistical difference, the serious complications (Clavien–Dindo, CD ≥ 3) in the NAC plus liver resection was almost seven times (10.1% vs. 1.6%) higher than that in the RFA group. The length of hospital stays (LOS) was significantly superior in the RFA group. The LOS in liver resection group was 3 times than the RFA group (7 days vs. 2 days, *p* < 0.001). Similarly, in terms of intraoperative blood transfusion, RFA also showed a significant advantage than liver resection (0.0% vs. 8.5%, *p* = 0.044). No deaths were observed in either group of patients within 30 days after surgery.

### 3.3. Survival Analysis

All patients in the RFA group achieved effective ablation, and the liver resection group completed R0 resection. The median follow-up time of all enrolled patients was 34.5 months. The 1-, 3-, and 5-year LTPFS in the RFA group was 93.4%, 84.9%, and 84.9%, respectively.

Both before and after PSM, PFS in the NAC plus RFA group was improved compared to NAC plus liver resection (log-rank; *p* = 0.004 and *p* = 0.028). The 1-, 3-, and 5-year PFS in the liver resection and the RFA groups was 57.4% vs. 86.9% (*p* < 0.001), 38.8% vs. 55.3% (*p* = 0.035), and 18.8% vs. 32.0% (*p* = 0.255), respectively. After PSM, the 1-, 3-, and 5-year PFS in the liver resection and RFA groups was 51.5% vs. 85.4% (*p* < 0.001), 36.2% vs. 52.5% (*p* = 0.102), and 17.1% vs. 33.5% (*p* = 0.399), respectively (shown in Figure 2A,B). The multivariate results showed that poor PFS is independently related to the treatment modality of liver resection (hazard ratio (HR), 1.850; 95% confidence interval (CI): 1.248–2.743; *p* = 0.002), CRS > 2 (HR, 1.555; 95% CI: 1.087–2.225; *p* = 0.016), and no response (stable disease (SD) + progressive disease (PD)) to NAC (HR, 1.643; 95% CI: 1.135–2.379; *p* = 0.009) (Table 4).

For OS, there was no statistical difference (log-rank; *p* = 0.545 and *p* = 0.885). The 1-, 3-, and 5-year OS in the liver resection and RFA groups was 100% vs. 96.7% (*p* = 0.191), 73.8% vs. 73.6% (*p* = 0.660), and 50.5% vs. 46.2% (*p* = 0.212), respectively. After PSM, 1-, 3-, and 5-year OS was 100% vs. 95.8% (*p* = 0.475), 67.9% vs. 70.4% (*p* = 0.811), and 43.4% vs. 47.1% (*p* = 0.399), respectively (shown in Figure 3A,B). Cox regression analysis also showed no difference in OS between the RFA and liver resection groups (Table 5).

## 4. Discussion

The current treatment options of locally curable CRLM are a comprehensive approach guided by MDT [26,27] rather than RFA or liver resection or NAC alone. Global guidelines or consensuses recommend NAC plus liver resection for CRLM, which is the current first-line treatment option for resectable CRLM [10,11,12,13]. Clinically, most CRLM patients are elderly. Hepatectomy is not suitable for all patients due to the large trauma, higher postoperative complication rate, and prolonged recovery. RFA is a commonly used minimally invasive treatment for liver tumors. RFA has numerous advantages, such as less trauma, fewer complications, and shorter hospital stay. Therefore, it is widely used in the treatment of liver tumors [28,29]. However, in most cases, RFA alone has a worse prognosis than liver resection alone in CRLM patients [30,31]. Therefore, there is an urgent need to propose a new therapeutic modality for CRLM patients who require local ablation. NAC improves the PFS of CRLM patients [8,32,33]. In addition, NAC followed by RFA has shown prognostic advantages in CRLM [21,22,34]. However, the choice between liver resection and RFA after NAC has not yet been studied. Therefore, the present study aimed to investigate whether NAC followed by RFA can replace NAC followed by liver resection.

This retrospective study mainly assessed the difference in postoperative data and prognosis between NAC followed by RFA and NAC followed by liver resection (defined as the RFA and liver resection groups, respectively). In terms of OS, the RFA group was similar to the hepatectomy group. The PFS in the NAC plus RFA group was significantly improved compared to the NAC plus hepatectomy group. The liver resection group increased the risk of recurrence by 85% (HR = 1.850, *p* = 0.002). The PSM again reconfirmed the credibility of these findings. These analyses simulated the randomization of prospective studies, reducing bias due to confounding variables. To our knowledge, it is the first retrospective comparative research comparing the prognostic advantages of NAC followed by RFA vs. NAC followed by liver resection.

Only four basic clinical characteristics differed between the two groups in this study, which might have an impact on prognosis. T3-4 of CRC, synchronous liver metastases, and liver metastases greater than 30 mm were poor factors in CRLM [10,35]. The proportion of these baseline characteristics was slightly higher in the liver resection group than in the RFA group. Multivariate regression analysis of PFS or OS all showed these three bias factors had no significant effect for prognosis. Thus, the present study used PSM to remove bias.

The liver metastases of CRLM before NAC in the RFA group <3 cm accounted for 33%, while those >5 cm were 0. The high LTP rates are an obstacle to the widespread adoption of RFA [36,37]. Diameter is the important factor contributing to LTP of RFA [38,39]. The efficacy of RFA decreases with increasing lesion size. The 1-year LTPFS in this study was significantly better than RFA alone (93.4% vs. 67.4–85%) [35,36]. This might be due to the shrinkage of liver metastases and tumor necrosis after NAC. However, the specific mechanism remains to be explored.

Strikingly, the PFS of NAC followed by RFA was better than NAC followed by liver resection. Most of the literature suggests that compared with RFA alone, liver resection alone can achieve better prognosis. The best results were similar outcomes in the two groups [40,41,42]. In multivariate analysis, NAC followed by RFA improved PFS, which could be attributed to the following reasons: NAC decreased the diameter of liver metastases; the micrometastases sites were eliminated, and the local and distant recurrence rates were decreased; NAC causes tumor necrosis; RFA generates unexpected mechanisms after NAC (that need to be investigated further). Multivariate analysis also showed that CRS >2 is associated with poor PFS, which was proposed in the 1999 CRS [43]. The current guidelines and consensus recommend NAC for CRLM patients with CRS >2 [10,11]. Similarly, multivariate analysis results showed that the pathological reaction to NAC is associated with PFS, also confirmed in several studies. The pathological reaction to NAC is a strong predictor of the disease [44,45]. Therefore, during chemotherapy, the changes in liver metastases should be observed and the treatment strategy should be adjusted promptly. Combining guidelines and our center’s clinical experience, this study provides the indications for RFA following NAC [15,46,47]. (I) Unresectable liver metastasis resulting from progression after NAC; (II) intraoperative combined hepatectomy; (III) due to the combination of NAC, the diameter of liver metastases could be widened to about 5 cm, and the number of liver metastases could be widened to about 5–7; (IV) patient preferences; (V) World Health Organization (WHO) poor performance status.

NAC followed by RFA for CRLM is a new treatment modality benefiting the PFS. PSM or multivariate analysis confirmed that our new treatment modality has significant advantages in controlling disease progression. The benefit of PFS does not benefit the OS: 1. The patient died of non-tumor factors; 2. The treatment option for the patient after recurrence.

The perioperative advantage of NAC plus RFA can be predicted in terms of postoperative complication rate or length of hospital stay. High postoperative complication rates, especially severe complications, often affect subsequent treatment and long-term prognosis [48,49,50]. This might explain the poor PFS in the NAC followed by the liver resection group. Fewer complications and shorter hospital stay indicate less financial burden [51]. The comprehensive treatment modality of NAC followed by RFA is a cost-effective strategy. Liver injury during and after chemotherapy should be rigorously assessed [52,53]. How to balance the benefits and complications is the focus of attention in the future.

Nevertheless, the present study has some limitations. First, this is a retrospective single-center study. Several differences were detected in the clinical baseline characteristics between the two groups. Although PSM and multivariate analysis have addressed biases that may result from an imbalanced clinical baseline, PSM decreased the sample size. Second, the left and right location of colon cancer is related to prognosis [54]. Due to the sample size, this study did not conduct a stratified analysis. Third, RAS mutation status is a major prognostic tool in determining OS and PFS of CRLM [55,56,57]. Limited by retrospective studies, RAS and BRAF mutational status is not established routinely. Finally, some patients may be “initially locally curable”; however, further examination may demonstrate progression or complete remission after NAC, negating the need for local therapy. Due to the retrospective design, these patients could not be included in the present study. These deficiencies would be addressed by multicenter retrospective studies or prospective randomized controlled studies in the future.

## 5. Conclusions

The results of this study are promising and positive. NAC plus RFA is a minimally invasive and cost-effective way. This is a new first-line treatment option for CRLM patients requiring local therapy. In the future, several aspects need to be clarified further. For example, (1) Which basic indicators can be used to screen patients for individualized precision treatment; (2) What is the optimal interval between NAC and RFA? According to the characteristics of RFA, it may be <4 weeks (compared to liver resection); (3) Are targeted drugs needed to improve the objective response rate? (4) How many cycles of NAC are required? What is the best NAC regimen? To sum up, compared with NAC followed by hepatectomy, NAC followed by RFA is beneficial to control the disease progress, recover quickly and have fewer serious complications. The new model of NAC combined with RFA is currently in the preliminary research stage. This model will soon become a research trend. However, the application of this new treatment modality requires further validation in prospective clinical trials.

## Figures and Tables

**Figure 1 cancers-14-05320-f001:**
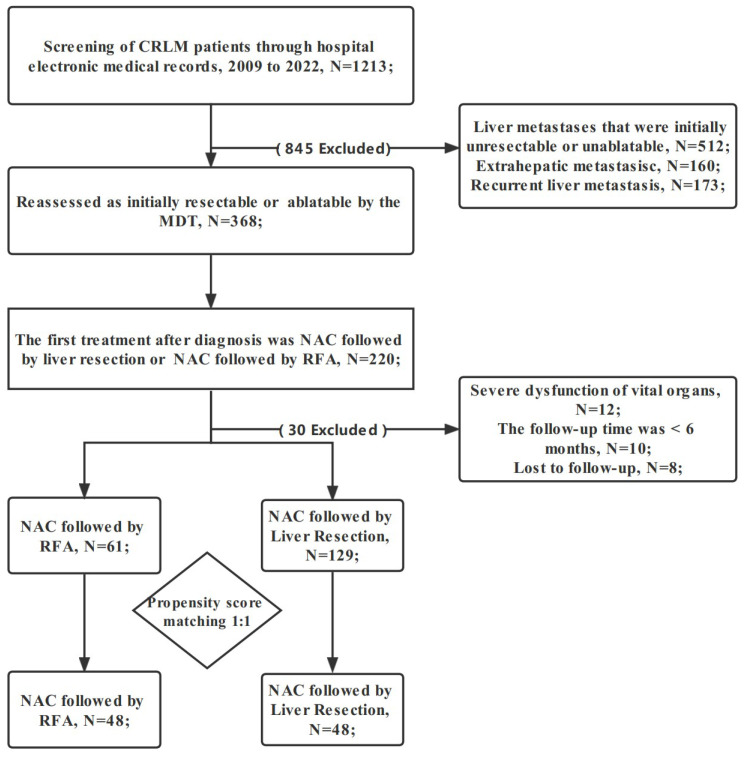
Flowchart of patient selection. CRLM: colorectal liver metastasis; RFA: radiofrequency ablation; NAC: neoadjuvant chemotherapy; MDT: multidisciplinary team.

**Figure 2 cancers-14-05320-f002:**
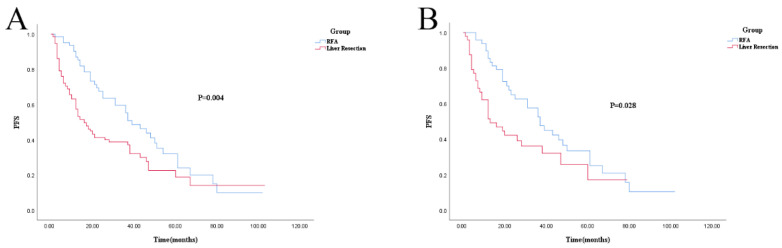
Kaplan–Meier survival curve for PFS of RFA and liver resection groups. (**A**): Unmatched analyses; (**B**): PSM analyses.

**Figure 3 cancers-14-05320-f003:**
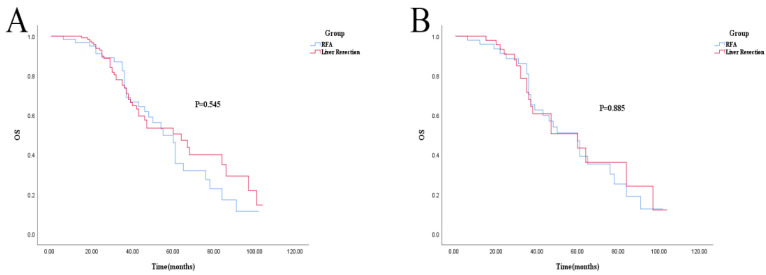
Kaplan–Meier survival curve for OS of RFA and liver resection groups. (**A**): Unmatched analyses; (**B**): PSM analyses.

**Table 1 cancers-14-05320-t001:** Baseline characteristics of the patients.

Variables	NAC plus RFA	NAC plus Liver Resection	*p* ^†^
N	61	129	
Age (years)			
≤60/>60	38/23	70/59	0.297
Gender			
Female/Male	27/34	32/97	0.007
CEA at diagnosis, ng/mL			
≤200/>200	54/7	122/7	0.136
Location of primary cancer			
Colon/Rectum	26/35	61/68	0.547
T stage of primary tumor			
T1-T2/T3-T4	9/52	7/122	0.031
N stage of primary tumor			
N0/N+	11/50	35/94	0.172
Timing of metastasis			
Metachronous/synchronous	23/38	30/99	0.038
Number of liver metastases			
=1/≥2	28/33	45/84	0.145
Largest diameter (cm)			
<3/≥3	41/20	60/69	0.008
Postoperative chemotherapy			
No/Yes	18/43	34/95	0.649
Response to NAC			
CR + PR/SD + PD	30/31	49/80	0.144
CRS ^‡^			
0–2/3–5	38/23	72/57	0.398

CEA: carcinoembryonic antigen; CR: complete response; PR: partial response; PD: progressive disease; SD: stable disease. ^†^: Fisher’s exact test, continuity correction and Pearson’s *x*^2^ test were used to analyze the basic characteristics. ^‡^: Node-positive CRC. Disease-free interval from CRC resection to the appearance of liver metastases of liver metastasis < 1 year. More than one metastasis. The largest liver metastasis > 50 mm. CEA in serum > 200 (mg/L).

**Table 2 cancers-14-05320-t002:** Baseline characteristics after PSM.

Variables	NAC plus RFA	NAC plus Liver Resection	*p* ^†^
N	48	48	
Age (years)			
≤60/>60	28/20	27/21	0.837
Gender			
Female/Male	17/31	17/31	1.000
CEA at diagnosis, ng/mL			
≤200/>200	41/7	44/4	0.336
Location of primary cancer			
Colon/Rectum	18/30	21/27	0.533
T stage of primary tumor			
T1–T2/T3–T4	3/45	3/45	1.000
N stage of primary tumor			
N0/N+	8/40	10/38	0.601
Timing of metastasis			
Metachronous/synchronous	14/34	14/34	1.000
Number of liver metastases			
=1/≥2	21/27	19/29	0.679
Largest diameter (cm)			
<3/≥3	31/17	31/17	1.000
Postoperative chemotherapy			
No/Yes	14/34	10/38	0.346
Response to NAC			
CR + PR/SD+PD	24/24	16/32	0.098
CRS ^‡^			
0–2/3–5	27/21	28/20	0.837

CEA: carcinoembryonic antigen; CR: complete response; PR: partial response; PD: progressive disease; SD: stable disease. ^†^: Fisher’s exact test, continuity correction and Pearson’s *x*^2^ test were used to analyze the basic characteristics. ^‡^: Node-positive CRC. Disease-free interval from CRC resection to the appearance of liver metastases of liver metastasis < 1 year. More than one metastasis. The largest liver metastasis > 50 mm. CEA in serum > 200 (mg/L).

**Table 3 cancers-14-05320-t003:** Perioperative conditions.

Variables	NAC plus RFA	NAC plus Liver Resection	*p* ^†^
N	61	129	
Overall complications, N (%)	9 (14.8%)	28(21.7%)	0.259
Abdominal infection, N (%)	6 (9.8%)	9 (7.0%)	0.693
Pleural effusion, N (%)	1 (1.6%)	10 (7.8%)	0.176
Liver failure, N (%)	2 (3.3%)	2 (1.6%)	0.815
Abdominal bleeding, N (%)	0 (0.0%)	6 (4.7%)	0.205
Serious complications (CD ≥ 3, N (%))	1 (1.6%)	13 (10.1%)	0.075
Length of hospital stay (median, days)	2 (1.2%)	7 (3.9%)	0.000
Intraoperative blood transfusion, N (%)	0 (0.0%)	11 (8.5%)	0.044

^†^: Fisher’s exact test or continuity correction or Pearson’s *x*^2^ or Wilcoxon rank sum test was used. CD: Clavien–Dindo classification.

**Table 4 cancers-14-05320-t004:** Analysis of prognostic factors associated with PFS.

Prognostic Factor	n	Univariate		Multivariate	
		HR (95% CI)	*p*	HR (95% CI)	*p*
Group					
RFA	61				
Liver resection	129	1.734 (1.179–2.550)	0.005	1.850 (1.248–2.743)	0.002
Gender					
Female	59				
Male	131	1.161 (0.790–1.705)	0.447		
Age (years)					
≤60	108				
>60	82	1.050 (0.731–1.509)	0.791		
CEA at diagnosis (ng/mL)					
≤200	176				
>200	14	1.056 (0.580–1.923)	0.858		
Primary tumor					
Rectum	103				
Colon	87	1.014 (0.709–1.450)	0.939		
T-stage of primary tumor					
T1/T2	16				
T3/T4	174	1.112 (0.581–2.127)	0.749		
LN metastasis					
No	46				
Yes	144	1.373 (0.871–2.164)	0.172		
Timing of metastasis					
Metachronous	53				
Synchronous	137	1.112 (0.750–1.650)	0.597		
Number of liver metastases					
<2	73				
≥2	117	1.361 (0.937–1.977)	0.105		
Size of largest lesion (cm)					
<3	101				
≥3	89	1.258 (0.881–1.797)	0.207		
Postoperative chemotherapy					
No	52				
Yes	138	0.888 (0.592–1.331)	0.566		
CRS					
1–2	110				
3–5	80	1.452 (1.018–2.070)	0.040	1.555 (1.087–2.225)	0.016
Response to NAC					
CR + PR	79				
PD + SD	111	1.593 (1.101–2.305)	0.013	1.643 (1.135–2.379)	0.009
Post-operation complications					
No	153				
Yes	37	1.208 (0.777–1.878)	0.402		

HR: hazard ratio; LN: lymph nodes; CEA: carcinoembryonic antigen; CEA: carcinoembryonic antigen; CR: complete response; PR: partial response; PD: progressive disease; SD: stable disease. CRS: Node-positive CRC. Disease-free interval from CRC resection to the appearance of liver metastases of liver metastasis < 1 year. More than one metastasis. The largest liver metastasis > 50 mm. CEA in serum > 200 (mg/L).

**Table 5 cancers-14-05320-t005:** Analysis of prognostic factors associated with OS.

Prognostic Factor	n	Univariate		Multivariate	
		HR (95% CI)	*p*	HR (95% CI)	*p*
Group					
RFA	61				
Liver Resection	129	0.868 (0.547–1.379)	0.549		
Gender					
Female	59				
Male	131	1.288 (0.787–2.110)	0.314		
Age (years)					
≤60	108				
>60	82	1.415 (0.890–2.249)	0.143		
CEA at diagnosis (ng/mL)					
≤200	176				
>200	14	0.753 (0.359–1.578)	0.452		
Primary tumor					
Rectum	103				
Colon	87	0.855 (0.541–1.351)	0.502		
T-stage of primary tumor					
T1/T2	16				
T3/T4	174	0.865 (0.374–1.999)	0.734		
LN metastasis					
No	46				
Yes	144	1.148 (0.628–2.097)	0.654		
Timing of metastasis					
Metachronous	53				
Synchronous	137	1.988 (1.177–3.359)	0.010	1.792 (0.994–3.231)	0.052
Number of liver metastases					
<2	73				
≥2	117	1.076 (0.680–1.702)	0.755		
Size of largest lesion (cm)					
<3	101				
≥3	89	1.181 (0.749–1.864)	0.474		
Postoperative chemotherapy					
No	52				
Yes	138	0.627 (0.387–1.017)	0.059	0.628 (0.387–1.021)	0.061
CRS					
1–2	110				
3–5	80	1.667 (1.057–2.630)	0.028	1.250 (0.748–2.089)	0.394
Response to NAC					
CR + PR	79				
PD + SD	111	1.023 (0.646–1.618)	0.924		
Postoperative complications					
No	153				
Yes	37	0.941 (0.525–1.685)	0.837		

HR: hazard ratio; LN: lymph nodes; CEA: carcinoembryonic antigen; CEA: carcinoembryonic antigen; CR: complete response; PR: partial response; PD: progressive disease; SD: stable disease. CRS: Node-positive CRC. Disease-free interval from CRC resection to the appearance of liver metastases of liver metastasis <1 year. More than one metastasis. The largest liver metastasis >50 mm. CEA in serum >200 (mg/L).

## Data Availability

Data for this study may be requested from the corresponding author where appropriate.

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
