# Peer review of "Neoadjuvant Chemotherapy Followed by Radiofrequency Ablation May Be a New Treatment Modality for Colorectal Liver Metastasis: A Propensity Score Matching Comparative Study"

_cancers, 2022, doi:10.3390/cancers14215320_

Round 1

Reviewer 1 Report

Dear authors

The work addresses a current and interesting topic. It comes to fill some answers regarding the approach to patients with colorectal liver metastases.

The work is well structured, but there are some items to be solved.

The most important problem, which can change the results of the study, is also found in the limitations of the study, meaning more homogeneity of the studied groups regarding the location of colon cancer (left or right), size of liver metastases, and last but not least numbers of used chemotherapy cycles and used NAC regimen.

As the chemotherapy regimen greatly influences prognosis, I believe that propensity score matching should also include NAC regimen, for more valuable study results, or like a less desirable option, this observation should be noted in the limitations.

As the chemotherapy regimen greatly influences prognosis, I believe propensity score matching should also include NAC regimen, for more valuable study outcomes, or as a less desirable option. This observation should be noted in limitations.

Also, please mention what were the criteria for choosing RFA or liver resection for the selected patients.

On the other hand, there are some minor observations:

Line 33: You have to mention what the abbreviation PFS stands for (progression-free survival);

Lines 107-109: There are some punctuation issues;

Line 125: A capital letter at the beginning of the sentence;

Table 3: Complete with % for percentages between brackets;

Line 267: "Surprisingly" is not necessary.

The patients with NAC followed by liver resection were selected from 2009 to 2022. The patients with NAC followed by RFA seem to be the same batch as the one in the paper cited in number 21. In the period June 2021-January 2022 there were no more patients with NAC followed by RFA eligible for study?

Patients with NAC followed by liver resection were selected from 2009 to 2022. Patients with NAC followed by RFA seem to be the same group as the one in the paper cited in number 21. Were no more patients with NAC followed by RFA eligible for the study during June 2021-January 2022?

Reviewer 2 Report

One of the first organ metastases of the colorectal cancers is the liver. In case of operability of the primary tumor, methodically there is also a chance for Neoadjuvant chemotherapy followed by the resection of the liver metastases, providing that the localisation of the metastatic tumors and the physical status of patients allows  this intervention.

Unfortunately, only about 20% of the patients are eligible for this treatment.  That was why the authors were looking for the utilization of another modality capable of replacing the liver resection. This was the previously also utilized radiofrequency ablation of the liver metastases, which exerts much less burden for the patients.

Thus they created two patient cohorts, where the two groups were harmonized by using the

propensity score matching (PSM). This method helped to create comparable cohorts, although the number of the participants decreased to 48 patients in both groups.

Next authors scrutinized the available clinical data, and their major goal was to compare the progression free and the overall survival of the two cohorts. They found that the new modality proved to be beneficial for PFS, but no difference was found in OS.  There were much more complications and longer hospitalization after liver resection. Although the manuscript does not emphasize, much more patients can be enrolled to the new therapy, where the patients do not need to endure the burden of liver resection.

In conclusion, authors created a new treatment modality, which could be utilized for many more patients with liver metastasis of colorectal cancer.  This so cold „in silico” scrutiny of their approach is summarized in the manuscript. Reviewer think that all kinds of therapeutic modality which offers a step forward in the management of cancer patients should be supported.

Reviewer is not an expert in statistical analyzes, thus this should be done by other reviewers.

Reviewer 3 Report

1. In the abstract section, line36, a p value of 0.075 is insignificant. The study claimed “RFA was significant lower than the liver resection (p=0.075)” is statistically big error.

2. Page 2of 16, line 53, “neoadjuvant chemotherapy is routinely used to prolong prognosis” Neoadjuvant is only used while down size and not necessary used. Also, how it can prolong prognosis? Can author cite the references? Most oncology studies as far I know, neoadjuvant chemotherapy is not has been shown to improve overall survival.

3. How to decide which patients should group to RFA treatment? There is clinical “equipoise” here. And I think the authors should make it clear. Did the clinician encourage for RFA? How did the patients make the choice? Is it due to economic cost for the patients or not? For surgery is only possible cure way.

4. Page 4 of 16, “at least one cycle of chemotherapy before local cure treatment is recognized as NAC.” One cycle of neoadjuvant chemotherapy is not the routine standard. One cycle of chemotherapy didn’t make big change. Can the authors illustrate an example that one cycle neoadjvuant really down size the metastasis sites and cite the reference?

5. In the methods part, the authors didn’t mention how did they use propensity score and matching for what?

6. Also, there is a meta-analysis regarding “Can Radiofrequency Ablation Replace Liver Resection for Solitary Colorectal Liver Metastasis? A Systemic Review and Meta-Analysis” It concluded that “The results showed that patients treated by liver resecton achieved better PFS and OS than those by RFA, but subgroup analysis and meta-regression displayed that the efficacy of RFA was equivalent to that of LR in solitary CRLM, when conditions were limited to tumors of ≤ 3 cm and fewer synchronous metastasis in the publication years 2011–2018. Meanwhile, RFA achieved lower complication rates when compared with liver resection. “ So I think it’s important for authors to select only those tumor ≦3cm after neoadjuvant chemotherapy for comparison and see if it really makes different.

è Overall, the designs with some methods defects and above mention points make the whole article less trustable.

Round 2

Reviewer 1 Report

Dear authors,

In this version, I consider the manuscript useful for readers/researchers.

Reviewer 3 Report

I cannot see how they correct the propensity score or redistribute it. Also, clinical "equipoise" is what I care most. Those tumor <=1cm could only be selected for RFA according to previous evidence and meta-analysis. Patient should be given clear information that those >1cm could only be cured by operation.